

# The WeIzmann Supercooled Droplets Observation (WISDOM) on a Microarray and application for ambient dust

Naama Reicher, Lior Segev, Yinon Rudich

Department of Earth and Planetary Sciences, The Weizmann Institute of Science, Rehovot, Israel

*Correspondence*: Naama Reicher (*naama.reicher@weizmann.ac.il*), Yinon Rudich (*yinon.rudich@weizmann.ac.il*)

**Abstract.** The WeIzmann Supercooled Droplets Observation on Microarray (WISDOM) is a new setup for studying ice nucleation in an array of monodisperse droplets for atmospheric implications. WISDOM combines microfluidics techniques for droplets production and a cryo-optic stage for observation and characterization of freezing events of

individual droplets. This setup is designed to explore heterogeneous ice nucleation in the immersion freezing mode, down to the homogenous freezing of water (235 K) in various cooling rates (typically 0.1-10 K min$^{-1}$). It can also be used for studying homogenous freezing of aqueous solutions in colder temperatures. Frozen fraction, ice nucleation active surface site (INAS) densities and freezing kinetics can be obtained from WISDOM measurements with excellent statistics of hundreds of individual droplets in a single freezing experiment. Droplets are surrounded by a mineral oil phase which assures the

isolation of the droplets and prevents mutual seeding, mass transfer or evaporation, and hence increases the reliability and reproducibility of the measurement. WISDOM also allows repeatable cycles of cooling and heating for the same array of droplets. This paper describes the WISDOM setup, its temperature calibration, validation experiments and measurement uncertainties. Finally, application of WISDOM to study the INP properties of size-selected ambient Saharan dust particles is presented.

**1 Introduction**

In mixed phase clouds, water droplets remain stable in a supercooled state below 273 K and ice nucleates spontaneously as droplets reach the homogenous freezing temperature, below 236 K (Pruppacher et al., 1998). At warmer temperatures, ice particles may coexist with supercooled droplets, due to heterogeneous nucleation facilitated by the presence of ice nuclei particles (INP)(Cantrell and Heymsfield, 2005). In cases where INP are immersed in the droplet before supercooling,

referred to as immersion freezing mechanism, the droplets first grow to supercritical size before freezing occurs (de Boer et al., 2011). Observations and modeling studies suggest that immersion freezing is the prominent mechanism for heterogeneous ice formation in mixed phase clouds (Ansmann et al., 2008; Field et al., 2012; Nagare et al., 2016; Possner et al., 2017; Rosenfeld and Woodley, 2000).



Ice particles affect the radiative and microphysical properties of mixed phase clouds and Earth's hydrological cycle. Therefore, they can influence present and possibly future climate (Hoose and Möhler, 2012; IPCC, 2013). Studying ice formation in clouds is hence important, and yet, due to its complexity, this process is still not fully understood and presents a great challenge to laboratory and field researchers as well as for clouds and climate modelers (DeMott et al., 2010; Schnaiter

et al., 2016; Ullrich et al., 2017).

Offline studies of immersion freezing often use cold stage techniques (Budke and Koop, 2015). The basic idea is to place an array of droplets over a cold stage and cool continuously until all are frozen, to obtain a quantitative measurement of their corresponding freezing temperatures (Vali, 1971). The droplets may be microliter-sized and observed with a simple camera. Smaller droplets, down to the pico-liter range, are usually observed under a microscope. In both cases, freezing events are

identified by optical changes in the droplets when they crystalize (Atkinson et al., 2013; Hiranuma et al., 2015; Knopf and Lopez, 2009; Murray et al., 2011).

Cold stage techniques may suffer from technical issues such as droplets evaporation and vapor transfer due to the Wegener–Bergeisen–Findeisen process, where ice grows on the expense of supercooled droplets or from seeding of neighboring droplets by formation and surface growth of frost halos (Budke and Koop, 2015). Some cold stages instruments place oil

over the droplets or use droplet in oil emulsions to prevent these effects (Murray et al., 2012). Still, results from cold stage experiments may be biased by effects of inhomogeneous temperature of the substrate and the surroundings or by various contaminations caused during droplets' preparation and measurement (Hiranuma et al., 2015). Furthermore, supercooling is limited due to the presence of impurities, which increases with the volume of the droplet. Hence, to allow comprehensive studies down to the homogenous region, low volumes (<1 μL) are used and generation of these volumes is not trivial and

may cause further complications.

Microfluidics is a technology of fluids manipulation in micro-channels array on a small device. Microfluidics is widely used in a range of fields, such as physics, chemistry, biology, life sciences and the food industry (Neethirajan et al., 2011; Sackmann et al., 2014; Whitesides, 2006). Recent studies used microfluidic apparatus to study ice nucleation processes. Riechers et al. (2013) used a microfluidics device to produce and collect monodisperse droplets of water in various sizes,

which were subsequently observed under a microscope to study their homogenous freezing. Stan et al. (2009) recorded nucleation in water droplets and silver iodide seeded droplets, while droplets were flowing during cooling. Schmitz et al. (2009) established 'Dropspots', a static microfluidic array of droplets, later used by Edd et al. (2009) to measure nucleation kinetics. However, in the atmospheric heterogeneous ice nucleation field, microfluidics techniques are not widely adopted, despite many potential advantages.

The WeIzmann Supercooled Droplets Observation on Microarray (WISDOM) is a new instrument that is combining the cold stage technique with microfluidics technology, and is designed to study immersion freezing of micrometer-sized droplets, while addressing most of the technical issues listed above. The WISDOM setup introduces several advantages of microfluidics to the atmospheric ice nucleation field. The microfluidic chips are cheap and easy to prepare and to operate. Production of droplets is fast and a range of droplet volumes can be used within the same microfluidic device. WISDOM  is





based on the 'Dropsopts' static array (Schmitz et al., 2009), which enables the separation and the fixation of the droplets, so that each individual droplet is recorded and studied, and also can be used for repetition of freezing cycles and further exploration of the nucleation process of a specific sample.

In this paper, we present the WISDOM setup, its calibration and validation procedures. For validation experiments,
homogenous and heterogeneous freezing were examined by following homogenous freezing rates of pure aqueous or solutions or deriving the efficiencies of three types of mineral dust surrogates and collected ambient Saharan dust to validate heterogeneous freezing experiments.

Homogenous nucleation rates are described stochastically using the volume-dependent ice nucleation rate ($J_v(T)$) in supercooled droplets, given by the frozen fraction ($f_{ice}$) of droplets with volume V at a certain temperature (T) and time
intervals ($\Delta t$) (Alpert et al., 2011; Murray et al., 2010; Riechers et al., 2013),

$$J_v(T) = \frac{-\ln\left(1 - f_{ice}(T)\right)}{V\Delta t},\quad\quad\quad\quad\quad\quad (1)$$

Heterogeneous freezing is described by a singular approach that assumes that nucleation occurs at a certain temperature due to special nucleation site. Hence, a cumulative number of nucleation sites per unit surface area, $n_s$, is used to describe the heterogeneous nucleation efficiency at a certain temperature,

$$n_s(T) = \frac{-\ln\left(1 - f_{ice}(T)\right)}{A},\quad\quad\quad\quad\quad\quad (2)$$

Where $f_{ice}$ is the fraction of frozen droplets at temperature T, and A is the specific surface area of the immersed particles in each droplet (Vali, 1971; Vali et al., 2015; Whale et al., 2015).

Validation of WISDOM was further extended below the homogenous nucleation temperature of pure water, using aqueous solutions as the freezing temperatures of solutions decrease as a function of the solution water activity.

## 2 Experimental setup

### 2.1 Droplets production and trapping

The WISDOM setup, shown in Figure 1, is made of a microfluidic setup which include a pressure controlled pump with 4 independent flow channels (OB1 MK3 by Elveflow), a stereoscope (SMZ-171 by Motic) that permits a full view of all channels and inlets, and a CCD camera (GS3 by Point Grey) that enable real time monitoring of the droplets production. The
flows in the channels are continuous and controlled by the pressure pump. One channel is connected to the continuous (oil) phase, and a second channel contains the sample (aqueous solution that can contain INPs). The two phases meet in a narrow junction where monodisperse droplets are generated due to the pressure exerted by one phase over the other. The ratio between the flows determines the size of the emerging droplets; the volume increases with increasing flow rate of the sample. In this setup, droplets are suspended in an oil mixture, consisting of mineral oil (Sigma Aldrich) and a 2 weight
percent (wt%) nonionic surfactant (span80®, Sigma Aldrich), added for droplets stabilization (Riechers et al., 2013). Hence, an array of picoliter (micrometer-size) droplets is generated directly on a device.



The principle of the Schmitz et al. (2009) design is that the droplets flow into round chambers that are connected by constriction channel. At a certain flow, the droplets are squeezed through the constriction channel and the array fills up with droplets. When the flow is too weak or stopped, the constriction channel stops the droplets' movement and they are trapped in the chambers. The droplets are isolated and stable in the chambers and it is safe to move the device from the generation

stage to the cold stage for the freezing experiments.

Devices are fabricated following the Schmitz et al. (2009) protocol. Briefly, the device pattern is imprinted on a polydimethylsiloxane (PDMS) polymer, later glued to a 1mm thick microscope glass slide using air plasma treatment. After the plasma treatment the PDMS surfaces are hydrophilic (Eddings, 2008). Therefore, the devices were used only in the following day, after their surfaces became hydrophobic, following their exposure to the atmosphere, or after their annealing

at 60°C for half an hour.

## 2.2 Freezing experiments and detection

The droplets array is placed in a commercial cryostage (Linkam, THMS600) coupled to an optical microscope (Olympus, BX-51 with 10X magnification, transmission mode). Experiments are monitored by a microscope mounted CCD camera (Allied Vision Technologies, Oscar F-510C) for automatic identification of droplets and their freezing events. Both the

device and the cooling stage are cleaned with 2-propanol. Then the device is placed over the stage together with a thin layer of oil on its bottom to provide good thermal conductivity. Each freezing experiment starts with dry $N_2$ purging to replace the moist atmosphere inside the cryostage to prevent condensation. During the experiment, $N_2$ flow prevents water condensation on the cryostage window. Freezing experiments are conducted with a cooling rate of 1 K min$^{-1}$ which is relevant for atmospheric conditions and also allows good thermalization of the droplets, as will be shown in the calibration section

(section 3.1). Each cooling cycle is followed by a heating cycle, where melting is observed. Analysis of the melting onset is then used to verify that the thermal conductivity is good and thus validate the measurement.

In-house LabVIEW software is used to record a freezing experiment movie file and analyze it offline. The temperature readings by the Linkam cryostage temperature sensor (<±0.25 K for the operated temperature range) and the movie frames are synchronized and integrated. In most cases, 1 second (or 0.017 K at 1 K min$^{-1}$) per frame is used. Currently, the

WISDOM setup operates with two types of devices that differ in their droplets' trap diameter: 40 μm and 100 μm. Approximately 550 and 120 droplets can be monitored per experiment in the smaller and larger diameter devices, respectively. Statistically, for the same sample, the larger droplets encompass more INP surface area within each droplet, which can be beneficial. The device can be reused for the same sample, if it is not clogged or destroyed during the experiment. However, because the channels of the 40 μm device are smaller they tend to clog faster (for instance by large

particles).



### 2.3 Automatic detection of phase transitions

The optical brightness of a droplet changes during a phase transition (freezing or melting) due to the different interaction of light with the liquid and the solids. For phase transition detection, an in-house image processing LabVIEW program monitors automatically the optical brightness change. The program detects the droplets using a spherical shape criterion and
sets a square surrounding the droplet that defines an array of pixels that are attributed to that specific droplet. A change in the optical brightness is represented by the gray level value of the image's pixels. Freezing is calculated per movie frame and is defined as the subtraction of the brightness mean value for each droplet in two consecutive frames, thus allowing derivation of freezing rates. At the beginning of the analysis, the first 15 frames are used to identify the noise level of the signal by calculating its standard deviation. The program then searches for the maximal freezing signal that is also greater than 5 times
the noise level. The temperature associated with this freezing signal is assigned as the freezing temperature for that droplet.

In this algorithm, the program can distinguish successfully between a phase transition event and noise that arises from the camera signal, droplet movement or any other interruption. Figure 2 presents a spectral analysis for different types of phase transitions observed in WISDOM. Since WISDOM operates in transmission microscopy mode, the light is scattered more efficiently by ice crystals in comparison with a liquid droplet and a freezing event involves droplet darkening and a negative
signal (Figure 2a). In comparison, during melting, the droplet becomes brighter until all the crystals melt and the signal is positive (Figure 2b+c).

## 3 Results and WISDOM validation

### 3.1 Temperature calibration

Temperature accuracy is a most important parameter in ice nucleation experiments. An error propagation analysis by
Riechers et al. (2013) demonstrated how the temperature uncertainty may lead to a distribution of temperatures between different instruments. Therefore we performed a thorough temperature calibration using the known eutectic melting points and the melting points of several aqueous solutions as calibration reference points. Although ice nucleation experiments are performed while cooling, the calibration experiments were done while heating to improve the calibration precision and to avoid biases associated with supercooling of the liquids (Budke and Koop, 2015).

### 3.1.1 Droplets thermalization

The temperature of the Linkam stage was measured at the upper center part of the cooling stage and hence may differ from the actual temperature of the droplets in the device due to thermal effects such as temperature gradients and temperature lag. During cooling or heating, a vertical temperature gradient may develop between the top of the device, in contact with the inner ambient of the cryostage, and the bottom of the device, which is in contact with the cooling silver block. This gradient
is expected to increase in magnitude, as the temperature of the stage decreases or increases below or above ambient



temperature. Edd et al. (2009) used a similar setup and found a difference between the top temperature and the bottom temperature of about 2 K around 237 K and 3 K around 227 K. Stan et al. (2009) also reported a vertical gradient of 1-2 K, that was reduced to 0.5 K with a flow of cooled $N_2$ over their device. In addition, a thermal lag may arise during cooling or heating as the rate of temperature change is high and precludes proper temperature equilibration. Hence, a more accurate

measurement of the droplet temperature is taken as a sum of the stage temperature with the contributions of both thermal gradient and lag.

Figure 3 demonstrates the combined effects of temperature change rate and device properties on the thermalization of pure water droplets (double distilled, 18.2 MΩ cm). Specifically, freezing and melting experiments at different rates were performed. The temperature difference (ΔT) is the difference between the measured values and the extrapolated temperature

at equilibrium conditions (0 K min⁻¹). As expected, at slower temperature cooling (heating) rates, the droplets are more equilibrated with the stage temperature and ΔT is negligible. However, ΔT increases at higher temperature cooling (heating) rates (e.g.; 10 K min⁻¹). We observed that during cooling (heating) the droplet is warmer (colder) than the stage and will freeze (melt) at colder (warmer) temperature at higher cooling (heating) rates. We also found that because ΔT is higher, in absolute value, for devices of thicker PDMS and/or in devices which hold larger droplets, it should be considered in the final

temperature calibration for these scenarios. Furthermore, ΔT was found to be almost symmetric for higher temperature cooling (heating) rates. However, for 1 K min⁻¹, ΔT during cooling is higher than that for heating. Our conjecture is that this can be an effect of the higher thermal gradient that develops as the temperature decreases well below ambient (236 K).

### 3.1.2 Melting of aqueous solutions

Figure 4 presents the measured melting points of NaCl solutions with different water activities. Reported melting points

represent the temperature in which all ice crystals in the droplets completely melted, in contrast with melting temperatures reported for pure liquids such as water, where the onset of melting is defined as the melting point. Melting temperature results were consistent with theoretical melting temperatures reported in Koop and Zobrist (2009). This provides support to our conclusion that droplets thermalize with the cooling stage when using a heating rate of 0.1-1 K min⁻¹. For faster heating rates (i.e. 10 K min⁻¹), the thermal lag was more pronounced, leading to a melting point shift of about 2-3 K. For more

concentrated solutions, faster heating rates shifted the melting points more.

### 3.1.3 Melting of eutectic solutions

Some aqueous solutions, such as NaCl and $MgCl_2$, arrange in a super-lattice at a certain wt% to form a solid with a well-defined melting point (eutectic) (252.05 K for NaCl and at 239.95 K for $MgCl_2$) (Borgognoni et al., 2009; Farnam et al., 2016). Interestingly, this type of melting has a smaller optical signature compared to that of melting points of pure substances, as can be seen in Figure 2b. We have set a specific water activity for a solution by determining its quantitative

composition using the extended aerosol inorganic model (E-AIM) (Clegg et al., 1998) at room temperature (298 K). For calibration purposes, because eutectic melting had a negligible variation for different water activities used in the range of




0.99 to 0.95, we decided to take their average to achieve a single melting value. These eutectic melting temperatures are colder than the melting point of pure water and, therefore, are used for expanding WISDOM calibration range.

The final calibration is obtained for a device with a specific PDMS thickness and at a specific cooling (heating) rate. For example, devices with 100 μm diameter sized droplet and of 4mm PDMS thickness have a linear calibration curve of

$T_{drop}=0.97*T_{stage}-0.46$ at 0.1 K min$^{-1}$.

### 3.2 Measurement reproducibility and device variability

Device's inter-variability was determined from 20 devices by comparing their corresponding homogenous freezing temperatures of pure water. Specifically, each device was recycled three times with freshly prepared droplets. Our results showed high reproducibility in the median freezing temperature, where 50% of the probed droplets froze ($T_{50}$), and high

reproducibility in the melting point temperature. Variation within the devices was always <±0.2 K at 1 K min$^{-1}$ and <±0.1 K at 0.1 K min$^{-1}$.

### 3.3 Homogenous freezing rates of pure water

Homogenous nucleation in supercooled water occurs in WISDOM between 238 and 237 K for a cooling rate of 1 K min$^{-1}$ and droplets diameter of 100 μm. Figure 5 shows WISDOM nucleation rates in comparison with other similar instruments. It

is seen that the slope of the rate and temperatures are similar to the slopes reported for other instruments. The temperature where 50% of droplets froze ($T_{50}$) is also in the expected range according to model results of Hoffer (1961). WISDOM rates are slightly slower, but within the uncertainty of the instruments used by Riechers et al. (2013) and Stan et al. (2009). Stöckel et al. (2005) show a higher nucleation rate. This discrepancy can be explained by a decrease in the number of surface nucleation events due to the oil phase surrounding our droplets, whereas in Stöckel et al. (2005), droplets are suspended in

air which allows surface nucleation may occur.

### 3.4 Homogenous and heterogeneous freezing of aqueous solutions

The water-activity-based ice nucleation theory by Koop et al. (2000) describes the dependence of the freezing temperature depression on the water activity ($a_w$) of the solution, regardless of the solute nature. Figure 6 presents the theoretical freezing and melting temperature curves from Koop et al. (2000) with homogenous ice nucleation results measured in WISDOM, for

four solutions with atmospheric relevance. Water activities for NaCl, ammonium sulfate (AS), glucose and levoglucosan mixtures were derived from the AIM model and were corrected for glucose and levoglucosan, for which water activity is temperature dependent (Knopf and Lopez, 2009; Zobrist et al., 2008). The experiments were conducted at 1 K min$^{-1}$ for 40 and 100 μm droplet diameters. The results follow the theoretical curves of the water-activity-based ice nucleation, and the dependence of the homogenous freezing on the droplet volume is as expected (Hoffer, 1961; Kuan-Ting and Wood, 2016) as

the curve of the smaller diameter droplets (green curve) is slightly colder compared with the larger volume droplets (dark green curve).





Similar experiments were conducted for 0.1 wt% of Arizona Test Dust particles (ATD, Powder Technology Inc.) immersed in glucose solution droplets. The ATD particles facilitate the ice nucleation at warmer temperatures, in agreement with similar studies (Hartmann et al., 2011; Niedermeier et al., 2010), and the freezing depression follow the water-activity-based ice nucleation curves. Here, the dependence of the freezing point on the droplet volume is more pronounced, as the surface

area of the immersed particles is higher, hence they contain higher number of nucleation sites (Marcolli et al., 2007) as will be shown in the next section for two more types of dust.

Below 223 K, ice nucleation occurs at slightly lower temperatures than expected by the theoretical freezing curve. As the WISDOM temperature calibration is not valid in this temperature range, we cannot conclude if this is due to a change of the thermal conductivity of the device or an effect of the high concentration of the solute in the water.

### 3.5 Heterogeneous nucleation and $n_s$ spectra of INP in pure water

#### 3.5.1 Standard dust powder

Heterogeneous freezing efficiencies of suspended mineral dusts K-Feldspar and Illite-NX in supercooled water droplets are presented in Figure 7 and summarized in Table 1, and are compared to recent published data. The particles are suspended at

different wt% and the frozen fraction of each suspension is derived as a function of temperature as represented by the color bar. To examine the freezing efficiency and compare the different mineral dust types, the results are normalized to the surface area within each droplet. Experiments were performed at 1 K min$^{-1}$ for 40 and 100 μm droplets diameters. Suspension preparation and evaluation of the surface area are described in the appendix.

The results demonstrate the effect of dust surface area immersed in the droplets on the freezing parameters. The freezing

temperatures increase with increasing surface area and are also reflected in the warming of the median frozen fraction ($T_{50}$) colored in yellow. The spectra of the number of nucleation sites per unit surface area ($n_s$) also support surface area dependence because all spectra converge to a single line. The $n_s$ results show the increase of nucleation sites at colder temperatures. Results from WISDOM are in good agreement with similar analyses from other instruments. In particular, $n_s$ is in best agreement with the Leeds-NIPI (Broadley et al., 2012; Murray et al., 2011) results both for K-Feldspar and for Illite-

NX particles. Results of Illite-NX particles are also in good agreement with the Binary instrument (Budke and Koop, 2015) and reside within the uncertainty of both instruments. The linear trend of few wt% support the assumption that particles in suspension are uniformly distributed and the droplets contain approximately the same surface area.

#### 3.5.2 Ambient mineral dust

Mineral dust particles were collected in Rehovot, Israel (31.9N, 34.8E about 80m AMSL), during dust storm events (DS#1

on the 9-11 March 2017, DS#2 on the 12-13 March 2017 and DS#3 on the 12-13 April 2017). The dust was transported from the Sahara Desert at all three events as presented in table 2. Size-segregated ambient dust particles were collected on





cyclopore polycarbonate filters using a Micro-orifice Uniform deposit Impactor (MOUDI; MSP Corporation model 110-R, (Marple et al., 1991)), that operated at 30 L min$^{-1}$ and for 24 hrs, similarly to Huffman et al. (2013) and Mason et al. (2015). MOUDI has eleven stages with cut points ($D_{50}$) of 0.056, 0.10, 0.18, 0.32, 0.56, 1.0, 1.8, 3.2, 5.6, 10, and 18 μm. The size distribution of the particles was obtained by Optical Particle Counter (OPC; GRIMM Technologies model 1.109) in the

range of 0.25-32 μm, and used for estimations of surface area immersed in the droplets (further details in Appendix B).

For heterogeneous freezing experiments, a quarter of each filter is placed with 300 μL DDW in 1.5 ml Eppendorf vial and particles were extracted by intensive dry sonication (Hielcher; model UP200St VialTweeter). In Figure 8, the spectra of the nucleation sites per unit surface area ($n_s$) of three super-micron stages ($D_{50}$ of 1.0, 1.8, 3.2 μm) are presented and summarized in T1. The freezing efficiency did not change significantly from one dust event to the other. It is also seen that

there are slightly more active sites for the larger particles (3.2 μm), as their surface area is higher and there is a higher chance to find an active site. In Figure 9, $n_s$ curves of the collected dust is compared to references of K-feldspar standard particles, analyzed in different instruments (the Leeds-NIPI (Atkinson et al., 2013), LACIS (Niedermeier et al., 2015)) and to measurements of ambient dust samples, from different locations around the world, including Israeli settled dust, that was analyzed in the AIDA chamber.  Moreover, the freezing of the size resolved mineral dust analyzed in this study by

WISDOM (slope in the temperature range) is consistent with the (green) polygon that portrays the estimated freezing efficiency by K-feldspar for natural concentrations of K-feldspar in internally mixed various mineral types (Atkinson et al., 2013). These results suggest that K-feldspar dominates the freezing of the super-micron ambient dust particles at temperatures 243-253K, and strengthens the hypothesis of Atkinson et al. (2013). It also seems that the Niemand et al. (2012) sample may have K-feldspar involved with the warmer part of their data (250-255 K), as their results also scale with

the Atkinson et al. (2013) scale for ambient samples.

### 3.6 WISDOM in comparison to other cold stage instruments

WISDOM was designed in order to overcome some of the technical issues and possible artefacts that cold stage techniques may suffer from. WISDOM's largest advantage is the use of microfluidics technology which solves some critical issues

inherent in other currently used instruments: (1) fast production of droplets minimizes sample sedimentation or other aging process that may occur in a suspension, leading to higher reliability of the measurements and to a better estimation of the surface area of the material that is exposed in the droplets. Moreover, several droplet diameters can be employed in the same device without its modification, (2) the high statistical power that can be achieved easily by fast analysis of thousands of droplets, (3) the droplets are monodisperse and individually analyzed, in contrast to other emulsion techniques as the

Differential Scanning Calorimeter (DSC) experiments, allowing to obtain the frozen fraction at each temperature, and hence to achieve detailed information about active sites and freezing rates that are important for atmospheric implications, (4) the use of oil minimizes possible artefacts from droplets' evaporation, neighbor seeding or vapor transfer due to the Wegener–



Bergeron–Findeisen processes, (5) the small droplet volumes decreases freezing artefacts by impurities, and in the absence of INPs the water freezes below the homogenous freezing threshold (-37ºC), in comparison with instruments that employ droplets with larger volumes which limit the workable temperature range, (6) using a static array opens the possibility to investigate several freezing cycles for the same droplets, (7) the microfluidics method and the small droplet volumes enable

working with small volumes which is an advantage when working with atmospheric samples.

WISDOM has a very accurate temperature calibration that spans a wide temperature range, using the eutectic freezing method. WISDOM most resembles the instrument used by in Edd et al. (2009). However, it seems that issues with temperature calibration in Edd et al. (2009) led to a temperature offset, and hence different freezing rates. Stan et al. (2009) achieved better temperature accuracy and high statistics. However, the freezing experiment was conducted in a flow more,

which is more complicated than in the WISDOM setup and needs complicated modeling. In addition, the cooling rates that were used were very fast, which induces additional errors. Riechers et al. (2013) had high temperature accuracy as they also used a DSC. However, they had to collect the droplets from the device as there was no static array option and this may add further complication and contamination.

Despite these advantages, WISDOM has a few disadvantages due to the use of microfluidics technology. These may include:

(1) several studies have shown that the oil may interact with some of the analyzed particles, possibly leading to biased data, (2) the microchannels are very small and there is risk for channels blockage and subsequently particles' filtration. This may lead to an error in the nucleation rates and efficiency that cannot be assessed properly, (3) it is not possible to perform any post analysis to the droplets content after the experiment.

## 4 Summary and conclusions

The new setup WISDOM is based on microfluidics technology and its detailed validation is presented. Based on a set of validation measurements and a good agreement with other instruments, we conclude that WISDOM is a suitable tool for studying atmospheric ice nucleation, both in homogenous and heterogeneous immersion freezing modes. Results of homogenous freezing correspond to water-activity-based nucleation theory in supercooled droplets and represent well volume nucleation rates. Heterogeneous freezing in supercooled droplets also agrees well with literature data. Furthermore,

freezing efficiency dependence on the particles surface area within the droplets is clearly observed. Monodisperse droplets are easily prepared from low volumes of sample. The microfluidic chips are cheap and easy to prepare and to recycle. The fast production of droplets reduces possible sedimentation or aging processes that may occur in the suspension. WISDOM also provides high statistics for each experiment, hence it can be used for studies of field samples as well, and cooling down to homogenous temperature range allows an extensive investigation of atmospheric particles nucleation sites. In this work

we have also demonstrated how WISDOM can be applied for studying the ice nucleation properties of ambient samples that contain very small quantity of sample. The particles were collected using the MOUDI during Saharan dust storm events.



Results are in correspondence with Atkinson et al. (2013) and demonstrate the importance of K-feldspar for ice nucleation in clouds, but further analysis of the mineralogy is still needed in order to verify that.

The authors declare that they have no conflict of interest.

*Acknowledgements*. We gratefully acknowledge support from the Ice Nuclei Research Unit (INUIT) of the German DFG, to The Helen Kimmel Center for Planetary Sciences, to the De Botton Center for Marine Sciences and for the Weizmann – UK Making Connections Program for funding this work. We also thank Prof. Daniel Knopf, Dr. Carsten Budke, Prof. Thomas Koop, Prof. Ido Braslavski, and Prof. Nir Freidman for their advices and to Prof. Ben Murray and Dr. Heinz Bingemer for
sharing K-feldspar and Illite-NX powder standards.

**Appendix A: Suspension preparation and characterization**

Illite-NX, ATD and K-feldspar powders were suspended in double distilled water and sonicated twice for 30 seconds with a 20 seconds pause, using Hielcher up200St VialTweeter, adjusted especially for Eppendorf vials. K-feldspar suspensions were additionally stirred overnight as sonication alone was not enough to achieve a good suspension and intensive
sedimentation was observed. For validation experiments, suspensions of 0.1 to 1 wt% were used. Characterization of the powders can be found in Hiranuma et al. (2015), Marcolli et al. (2007) and in Atkinson et al. (2013) and quantification the powders specific surface area was based on $N_2$ adsorption analysis of Brunauer–Emmett–Teller (BET) (Brunauer et al., 1938) using Quantachrome Instruments Nova 2200e and resulted in $1.9\pm0.6$ m$^2$ g$^{-1}$ for the K-feldspar powder, $108.6\pm2.8$ m$^2$ g$^{-1}$ for the Illite-NX powder and $37.1\pm1.4$ m$^2$ g$^{-1}$ for the ATD powder. In order to ensure a proper analysis of the surface area,
and avoid possible surface contaminants as water, surface cleaning was done by degassing the powders at 60K for 3 hours ahead of the BET analysis. Evaluation of the surface area in each droplet was then calculated by the wt% which was used, knowing the approximate surface area per mass and assuming that the mass is distributed uniformly inside the droplet with the same volume. Error in $n_s$ is then propagated by the error of the total surface area (4% for ATD, 2.6% for Illite-NX and 30% for K-feldspar), by the error in the droplet volume assessment (3-5% for 100 μm droplets and 6-8% for 40 μm droplets)
and by the error in the frozen fraction (15%) resulting in 17% for Illite-NX and ATD measurements and 35% for K-feldspar measurements.

**Appendix B: Collection of ambient particles during dust storm events in Rehovot**

GRIMM measurement was synchronized to the MOUDI stages for the estimation of the total surface area that was collected on the filter for droplets surface area estimation. For that, two base assumptions were made: (1) all the particles that were
collected are extracted to the water later used for the freezing experiments, (2) sphericity of the particles. The GRIMM bins





are synchronized to the MOUDI stages based on collection efficiency of the MOUDI, obtained from Marple et al. (1991). For example, on certain MOUDI stage, all the particles that own diameter that is larger than the $D_{50}$ have high chance to be impacted on that stage. All the rest of the sizes, that are smaller in their diameter, will continue to the next stage and will have high chance to deposit there. Hence, the GRIMM's bins were synchronized to the MOUDI $D_{50}$ stages. For the $n_s$

calculations, the surface area was based on the number of particles that were measured in a certain bin and their total surface area. To calculate the surface area of a particle (assuming sphericity) in a certain bin, the midpoint of that bin was used as a radius. To calculate the total mass of the particles in each filter, dust density of Quartz was used (2.65 g cm$^{-3}$), as this is usually the dominant mineral (Mahowald et al., 2014).

For control, analysis of blank filter was done. The blanks were sonicated before analysing them and freezing was mostly

colder than the freezing temperatures that are presented here and hence no special reduction of the final active sites was done. Error of 30-40% in the $n_s$ value is derived from the estimation of the surface area in the droplets due to the OPC bin size.

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

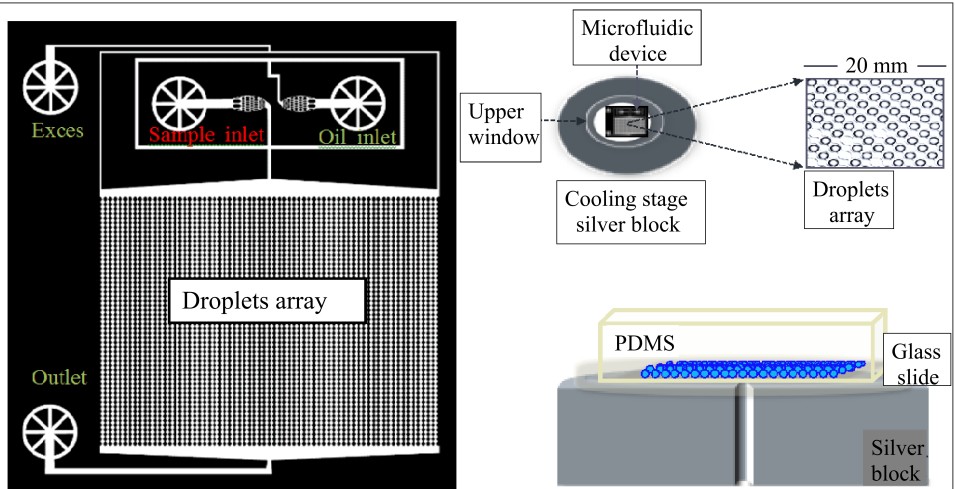

**Figure 1:** The WISDOM setup. a) The design of the microfluidic device is based on Schmitz et al. (2009). Aqueous solutions (including the sample) and oil are connected through the inlets and merge in a junction to generate monodisperse droplets. Subsequently, droplets flow into a trap array and settle in them as the flow is stopped. The device is transferred into a cooling stage for subsequent freezing
10   experiments. b) upper and c) side views of the device, which is made of PDMS, plasma glued to a microscope glass slide, placed over the cooling silver block.



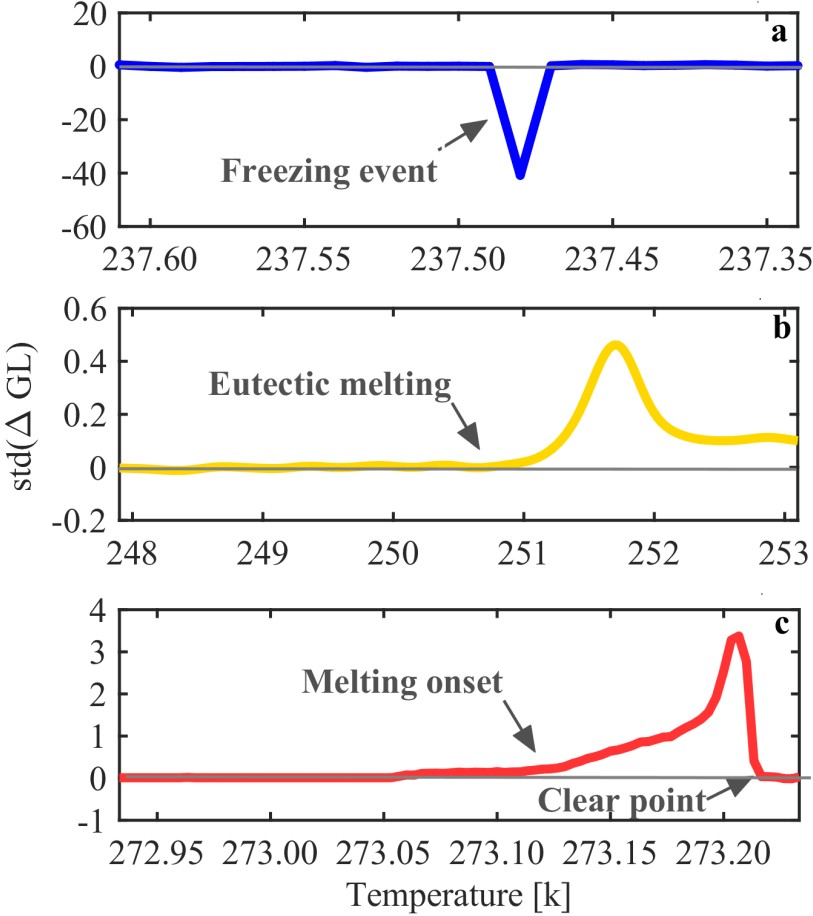

**Figure 2:** Spectra of different phase transition events as observed in WISDOM. a) freezing, b) eutectic melting, and c) melting onset and clear point (liquefaction). The phase transition is defined optically by the brightness information obtained by the gray level of the image pixels. Freezing and melting examples are for pure water droplets and the eutectic melting example is for aqueous solution droplets of NaCl. In all cases the droplets diameter is 100 μm.



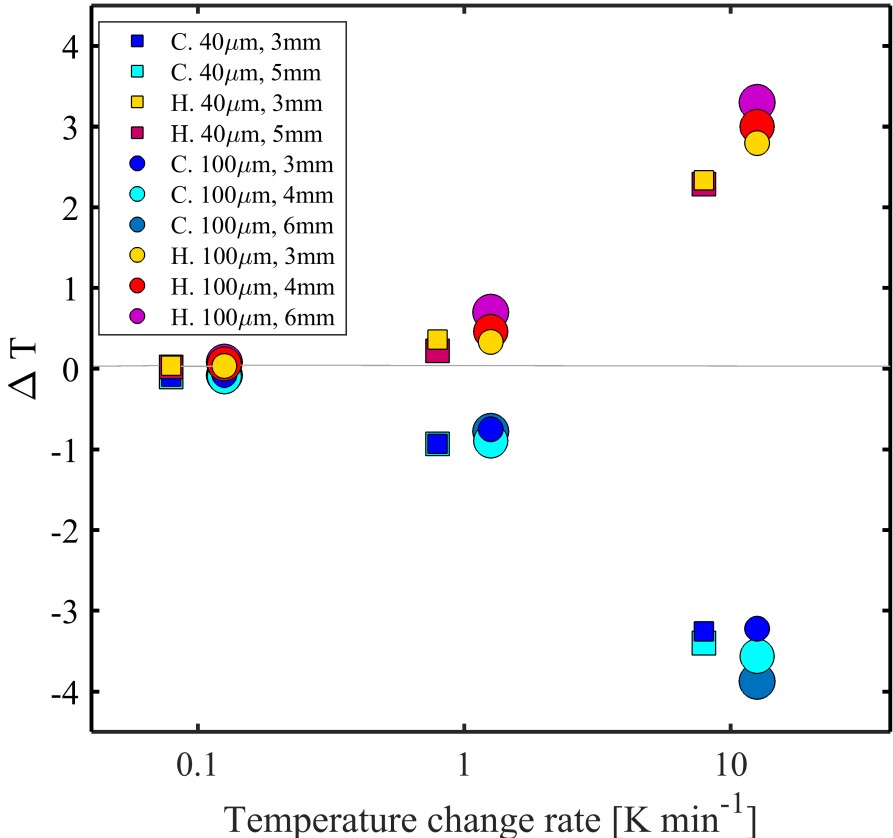

**Figure 3.** The temperature difference (ΔT), defined as the temperature difference between the stage temperature and the droplet extrapolated temperature at equilibrium conditions at different cooling (heating) rates. Freezing and melting points of pure water are represented by circles and squares (40 and 100 μm droplet diameter, in correspondence) for different PDMS thicknesses and are represented by different colors. Droplets are close to equilibrium with the stage temperature at rates <0.1 K min⁻¹ and ΔT increases with increasing temperature change rate and with the PDMS height.





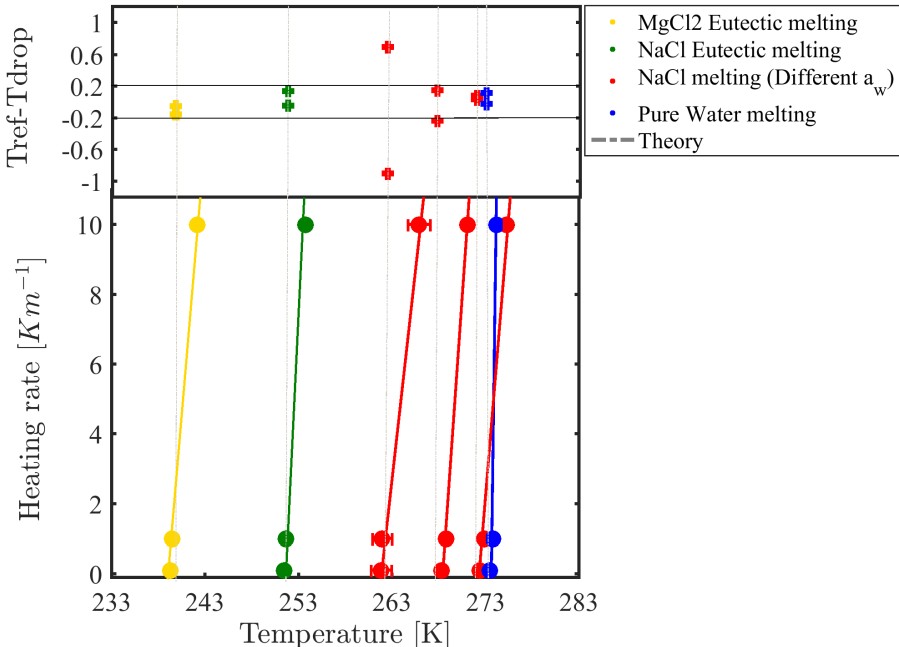

**Figure 4.** Temperature calibration by melting points of eutectic solutions and pure water droplets for different heating rates. Calibration is presented for 100 μm droplets with 4 mm PDMS thickness. The onsets of pure water droplets are also taken into account. Eutectic melting is used for the colder temperatures range (<253 K) while clear point (liquefaction) at various water activities is taken for the warmer temperature range. The upper panel presents the temperature difference between the reference value and the cooling stage temperature after calibration. Most of the differences are within the range ±0.2 K.





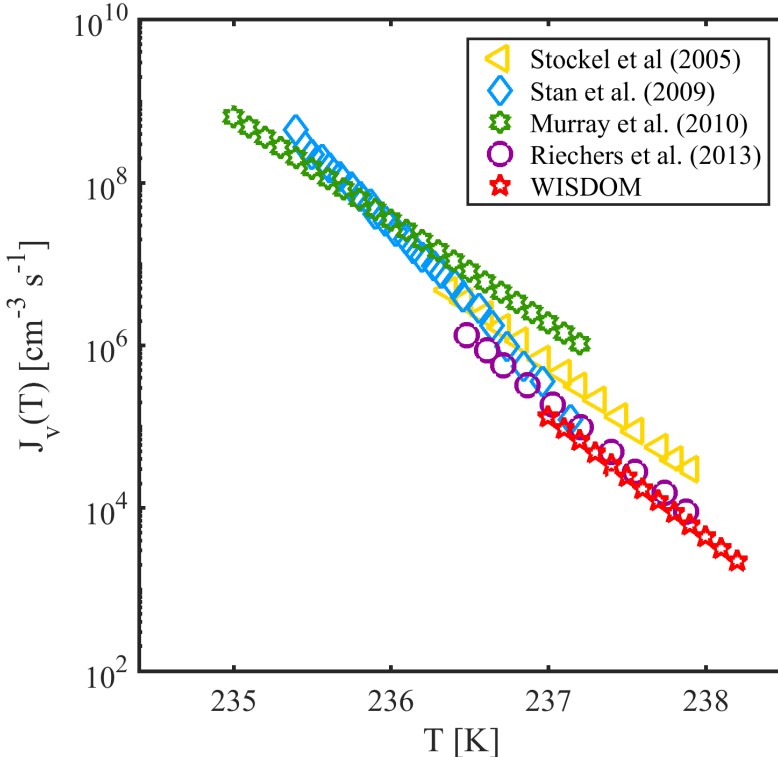

**Figure 5.** The volume-dependent homogenous freezing of pure water, derived for 100 μm droplets with 4 mm PDMs height. WISDOM rates are compared to relevant literature data. The obtained fit from WISDOM is $J_{v(T)}=\exp(-3.4T+817.6)$. Temperature uncertainty for WISDOM is ±0.25 K.





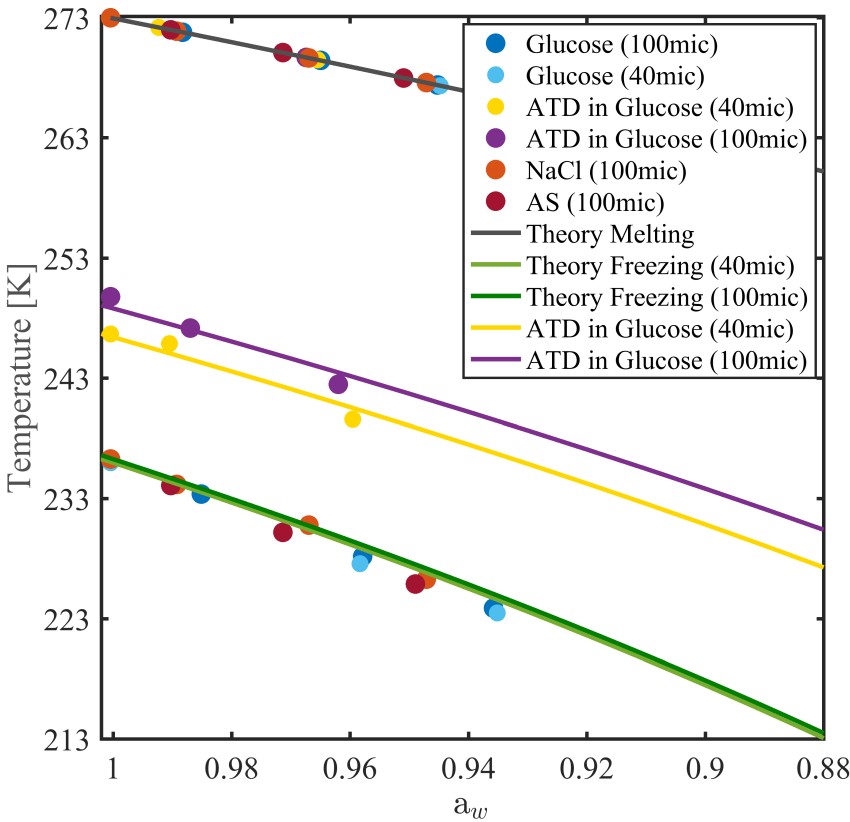

**Figure 6.** Homogenous and heterogeneous ice nucleation temperatures for 40 and 100 μm aqueous solution droplets as a function of solution water activity. Freezing and melting curves are derived from Koop et al. (2000). Heterogeneous ice nucleation is performed with 0.1 wt % ATD particles immersed in the droplets.





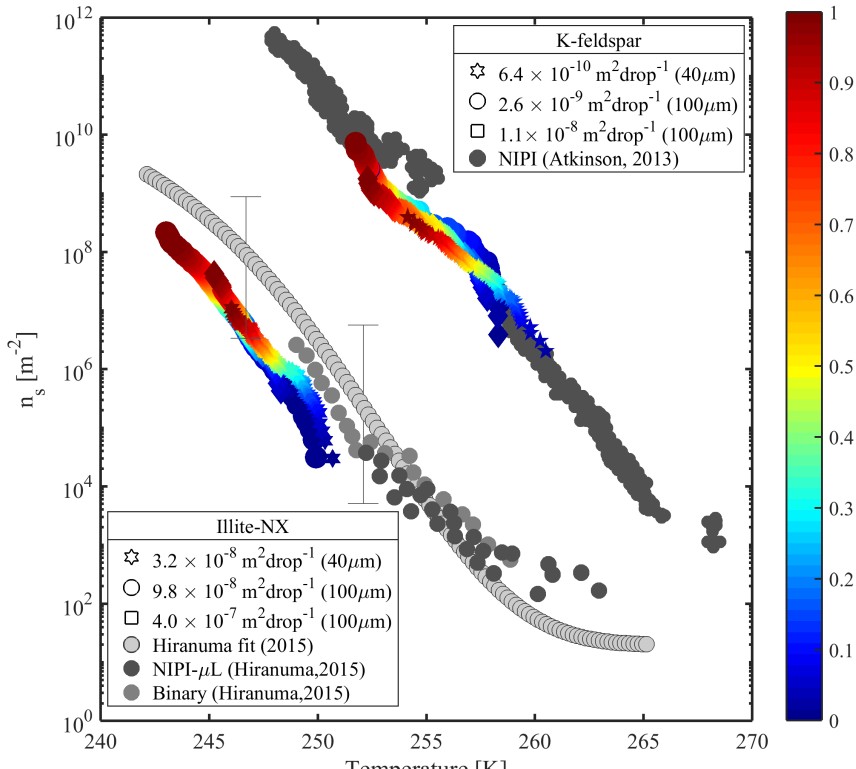

**Figure 7.** Accumulated active site density spectra ($n_s$) of K-feldspar and Illite-NX particles as a function of temperature from validation experiments of immersion freezing in WISDOM. Frozen fraction values are represented by a color bar, for few surface area values that are exposed in 40 and 100 μm droplets. The dependence of the nucleation site density on the surface area is illustrated here. For validation, previous immersion freezing measurements are also presented (Hiranuma et al. (2015) and Atkinson et al. (2013)). For the Hiranuma et. al. fit, the maximum deviation between maxima and minima in the vertical axis are shown by the error bars for the relevant temperature range.





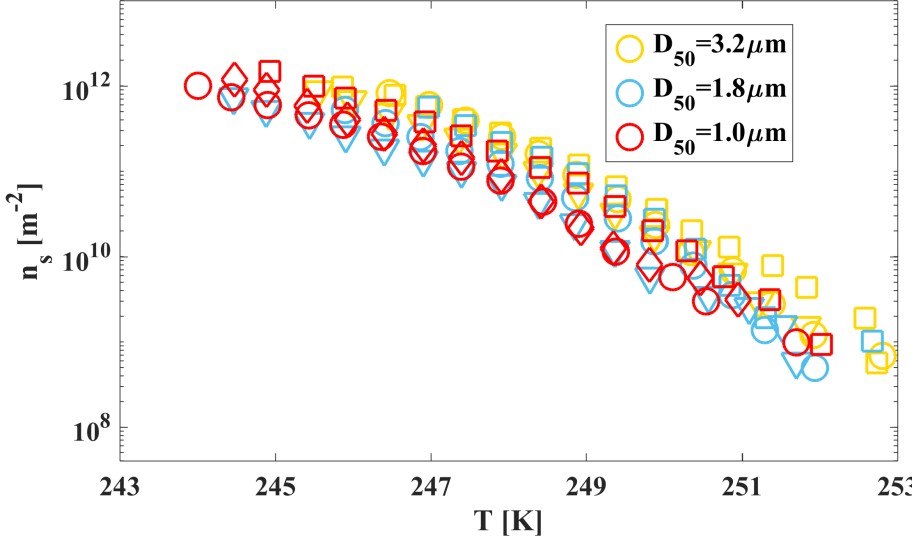

**Figure 8.** Accumulated active site density spectra ($n_s$) of ambient super-micron mineral dust particles collected in Israel during three dust events in 2017, for three different sampling stages of the MOUDI; $D_{50}$ of 1, 1.8 and 3.2 µm. DS#1 marked by circles, DS#2 marked by squares and DS#3 marked by triangles.





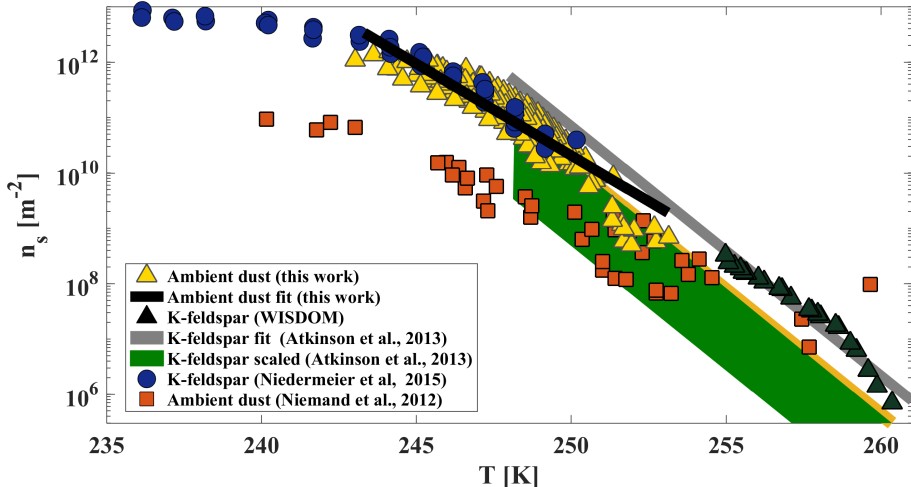

**Figure 9.** Accumulated active site density spectra ($n_s$) of ambient super-micron mineral dust particles collected in Israel during three dust events in 2017 for the three dust storms and for the three MOUDI stages that were analyzed with $D_{50}$ of 1, 1.8 and 3.2 µm. The fit $ln(n_s) = -0.763T + 214.5$ is also presented. References of K-feldspar standard particles activated in WISDOM, Leeds-NIPI (Atkinson et al., 2013) and LACIS (Niedermeier et al., 2015) instruments are presented, as well as ambient dust particles that were analyzed in AIDA and included Israeli dust (Niemand et al., 2012).



**Table 1.  Summary of immersion freezing experiments performed for WISDOM validation.**

| | Droplets diameter [μm] | SA [m² drop⁻¹] | $T_{50}$ [K] | BET [m²g⁻¹] |
|---|---|---|---|---|
| **Illite-NX** | | | | |
| 0.2wt% | 95.1±3.6 | $3.2 \times 10^{-10}$ | 246.4 | |
| 0.8wt% | 96.1±2.9 | $9.8 \times 10^{-08}$ | 247.8 | 108.6±2.8 |
| 1wt% | 38.2±2.4 | $4.0 \times 10^{-07}$ | 245.4 | |
| **K-feldspar** | | | | |
| 0.2wt% | 99.6±2.8 | $6.4 \times 10^{-10}$ | 255.3 | |
| 0.8wt% | 98.2±2.6 | $2.6 \times 10^{-09}$ | 257.0 | 1.9±0.6 |
| 1wt% | 39.8±2.4 | $1.1 \times 10^{-08}$ | 253.3 | |
| **0.1 wt% ATD** | | | | |
| **in glucose** | | | | |
| $a_{w=1}$ | 98.1±3.8 | $1.8 \times 10^{-08}$ | 250.0 | |
| $a_{w=0.987}$ | 101.2±2.9 | $2.0 \times 10^{-08}$ | 247.5 | |
| $a_{w=0.962}$ | 99.1±4.6 | $1.9 \times 10^{-08}$ | 242.9 | 37.1±1.4 |
| $a_{w=1}$ | 38.3±3.2 | $1.1 \times 10^{-09}$ | 246.2 | |
| $a_{w=0.991}$ | 39.9±3.3 | $1.2 \times 10^{-09}$ | 240.2 | |
| $a_{w=0.959}$ | 37.3±2.8 | $1.0 \times 10^{-09}$ | 236.3 | |
| **DS#1** | | | | |
| **09-11/03/17** | | | | |
| $D_{50}$   1.0 | 96.3±6.8 | $4.1 \times 10^{-12}$ | 246.7 | - |
| [μm]   1.8 | 97.3±4.2 | $8.1 \times 10^{-12}$ | 248.3 | |
| 3.2 | 94.7±7.9 | $5.8 \times 10^{-12}$ | 248.2 | |
| **DS#2** | | | | |
| **12-13/03/17** | | | | |
| $D_{50}$   1.0 | 89.9±4.3 | $3.2 \times 10^{-12}$ | 247.5 | - |
| [μm]   1.8 | 95.6±9.6 | $7.8 \times 10^{-12}$ | 248.8 | |
| 3.2 | 89.4±10.8 | $5.3 \times 10^{-12}$ | 248.7 | |
| **DS#3** | | | | |
| **12-13/04/17** | | | | |
| $D_{50}$   1.0 | 97.3±3.8 | $2.9 \times 10^{-12}$ | 246.5 | - |
| [μm]   1.8 | 92.1±4.7 | $5.1 \times 10^{-12}$ | 246.8 | |
| 3.2 | 96.9±6.9 | $5.4 \times 10^{-12}$ | 248.1 | |



**Table 2. Summary of dust event over Israel that were sampled for WISDOM validation.**

| Dust event: | DS#1 | DS#2 | DS#3 |
|---|---|---|---|
| Date: | 9-11 March 2017 | 12-13 March 2017 | 12-13 April 2017 |
| NOAA HYSPLIT MODEL Backward trajectories | | | |
| | | | |