# Peer review of "The Welzmann Supercooled Droplets Observation (WISDOM) on a"

_Atmospheric Measurement Techniques, 2017_

## Referee Comment (RC1) · Anonymous Referee #1 · 2 Oct 2017

Reicher et al. describe the calibration and operation of a cold-stage for measurement of ice nucleation activity. The instrument used a microfluidic device for generation of monodisperse droplets inside an oil phase. The device is operated in batch mode. Droplets are prepared in a continuous flow system. Then the flow is stopped and the device is placed on a cold-stage/microscope for nucleation experiments. Characterization experiments using pure water, eutectic solutions, and several test dusts are described. Application of the technique to ambient aerosol is demonstrated.

This is a solid paper that describes the application of a microfluidic device for ice nucleation studies. The characterization and validation experiments are of high qual-

ity. Specifically, the temperature characterization and comparison with the various test dusts at water activity of one and less than unity are thorough and convincing. The main conclusion of the paper that "WISDOM is a suitable tool for studying atmospheric ice nucleation, both in homogeneous and heterogeneous immersion freezing" is well justified.

However, the paper should be viewed in perspective of the current literature. At least 10+ similar cold-stages have been built, characterized, and described in the last few years. This is also not the first cold-stage that uses a microfluidic device. In fact the predecessor work by Stan et al. (2009) managed to operate the instrument in continuous flow mode, rather than batch mode, and thus is more technologically advanced than the work here.

In many places the manuscript tries to 'sell' or 'justify' the microfluidic technique presented here as an advance in technological capabilities. For examples, the microfluidic technique is 'cheap', 'easy to prepare', 'easy to operate', 'production of droplets is fast', 'a range of droplet volumes can be used', 'cooling down to homogenous temperature range allows an extensive investigation of atmospheric particles'. A list is provided that suggests that WISDOM solves 'some critical issues inherent in other currently used instruments', including 'fast production of droplets', high statistical power' due to 'fast analysis of thousands of droplets', 'droplets are monodisperse and individually analyzed', 'the use of oil minimizes possible artefacts', 'small droplet volumes decreases freezing artefactsby impurities, 'static array opens the possibility to investigate several freezing cycles for the same droplets', and 'microfluidics method and the small droplet volumes enable working with small volumes'.

The statements above are either misleading or wrong or have not been demonstrated in this paper (see further below). It indicates that the authors have not critically reflected on the differences and similarity of WISDOM with the current technology and some of the necessary tradeoffs that are being made when designing cold stages. The WISDOM instrument does not demonstrate any aspects that has not been also addressed

in other designs. A blunt assessment of the technique is that it is on par with the current state of cold-stage designs. However, although the technique is certainly valuable (and cool!), it does not represent an advance in either science or technology relative to the published literature. A revised manuscript must better reflect the techniques ability and limitations relative to other existing techniques.

Issues with justification. Quotes from the manuscript are in bold font.

**The microfluidic chips are cheap and easy to prepare and to operate. Production of droplets is fast and a range of droplet volumes can be used within the same microfluidic device.**

These statements are highly subjective. Please elaborate. What does cheap mean? The chips are custom made, there is cost for that equipment and labor involved, which is not free. On top the system requires a microscope, liquid flow control, and more complex cooling setup relative to a regular cold-stage. What does fast mean? If cheap and fast are a 'selling' argument for the technique, then it should be compared to other cold-stage techniques. The real metric that need to be compared relative to other techniques

(1) Total cost of equipment, total cost per 'experiment' in terms of equipment and consumables.

(2) Number of drops and total volume that an be studied in a reasonable 'experiment' (e.g. working with a dust sample for one morning). Related to this number is the lowest and highest INP concentrations that can be captured in that experiment. Factors such as channel clogging, or chip damage due to expansion of water upon freezing should be accounted for here fairly.

(3) Ease of use relative to a regular cold-stage technique.

These are the key pieces of information that are needed to weigh whether one should adopt this technique or not (other than personal preference, which may well justify the

route of microfluidics).

**A range of droplet volumes can be used**

Other cold-stages have worked with the same range of droplet volumes and sample statistics.

**Cooling down to homogenous temperature range allows an extensive investigation of atmospheric particles**

**Furthermore, supercooling is limited due to the presence of impurities, which increases with the volume of the droplet. Hence, to allow comprehensive studies down to the homogeneous region, low volumes are used and generation of these volumes is not trivial and may cause further complications.**

The authors should elaborate on the point they are trying to convey here. Presumably, this is meant to convey that a technique like WISDOM is needed? Three points.

First, may authors have accomplished studying homogeneous freezing using small drops successfully in past and present studies. Clearly it is doable and the alluded to 'complications' have been solved in some way by these authors.

Second, one might be inclined to believe that going to small drop volume solves some problems. If done correctly, small volumes do allow generation of drops with no impurities and thus studying homogeneous nucleation (as demonstrated widely in the literature). However, the impurities are still present. Simply subdividing the sample into more droplets will not help raising the lower limit of detection of ice nucleation activity for that sample. Specifically, if a sample has $10^5$ nuclei per liter of liquid, studying a large number of small droplets or a smaller number of large droplets will produce the same result. This study has not demonstrated any advance in purifying water. The achievement of pure water freezing has been managed before by others when working with small volumes.

Third, the number of drops studied per experiment here is quite small ( 500). While this

means that homogeneous freezing temperature can be reached, it also means that the total volume studied is quite small, and thus the lower limit of detection for ice nuclei is much larger than in devices that use larger droplets.

**WISDOM solved some critical issues inherent in other currently used instruments including**

**(1) fast production of droplets minimizes sample sedimentation or other aging process that may occur in a suspension, leading to higher reliability of the measurements and to a better estimation of the surface area of the material that is exposed in the droplets.**

This has not been shown here. First, it is unclear how fast fast is. No times are given in the paper. Second, it has not been compared to how fast others can perform an experiment. Third, the paper does not demonstrate that this technique is more reliable than others. There is no metric for reliability. This statement is clearly not justified.

**(2) The high statistical power that can be achieved easily by fast analysis of thousands of droplets.**

Again, fast is subjective. The number of drops given here is that approximately 550 forty micron and 120 hundred micron droplets can be monitored per experiment. The duration of an experiment is unclear, but it includes chip production, chip loading, and post processing. So how many experiments can one person do in a week? Is the statistical power really higher than in other studies? For example Hader et al. (2014, ACP) generate and analyze 500-800 drops per experiment in the 80-100 micron size range. Another example Peckhaus et al. (2016, ACP) generate and analyze 1200-1500 drops drops per experiment om 100 micron size range. Both studies also use oil immersion. Does WISDOM have really faster analysis and higher statistical power than these studies? Perhaps so, but it must be proven.

**(3) the droplets are monodisperse and individually analyzed, in contrast to other**

**emulsion techniques**

See above for examples that also do individual analysis.

**(4) the use of oil minimizes possible artefacts from droplets' evaporation, neighbor seeding or vapor** . . .

See above cited studies and several other systems that do the same.

**(5) the small droplet volumes decreases freezing artefacts by impurities, and in the absence of INPs the water freezes below the homogenous freezing threshold (-37oC), in comparison with instruments that employ droplets with larger volumes which limit the workable temperature range.**

First, this point is incorrect as stated above. Working with small droplet volumes allows for study of dilute solutions and ice nuclei that have high concentration in liquid. The high concentrations are the result of the overall low liquid volume that is analyzed. Working with large droplets is a choice to increase the detection limit. Furthermore, many investigators worked with small droplets before, so this is not a distinguishing feature of WISDOM.

**(6) using a static array opens the possibility to investigate several freezing cycles for the same droplets**

Refreeze experiments have been performed with static droplet arrays since the pioneering work by Vali in the 1960's. This is not a distinguishing feature of WISDOM.

**(7) the microfluidics method and the small droplet volumes enables working with small volumes which is an advantage when working with atmospheric samples.**

First, working with small volumes and atmospheric samples has been also done by previous investigators. Second, it is incorrect to cite the small droplet volume as an advantage. It may be an advantage under some circumstances, but not others. Specifically, the poor lower limit of detection for small droplets is a clear disadvantage.

To summarize the critique of this section. The text is prefaced with "WISDOM's largest advantage is the use of microfluidics technology which solves some critical issues inherent in other currently used instruments". The issues that WISDOM solves have been solved in some form or another in previous studies. WISDOM shares the advantages and disadvantages of the design choices involving the selection of oil, droplet size, batch mode, and optical detection. Claims regarding high statistical power, ease of use, and cost need to be proven. The paper should give an honest assessment of whether a new investigator should pursue the microfluidics route or the setup or something else. WISDOM and the nanoliter droplet freezing assay have similar control over drop volume, automated drop generation, drop detection, refreeze capabilities, etc. Perhaps telling is that the entire section in the draft manuscript is devoid of any references to the literature to back up their claims.

Other comments

**However, in the atmospheric heterogeneous ice nucleation field, microfluidics techniques are not widely adopted, despite many potential advantages.**

What are these potential advantages? Please be specific. Also there are clearly potential disadvantages to microfluidics. These should be discussed as well.

Figure 2 needs better explanation. The y-axis is not clearly defined. Presumably, Delta GL stands for change in grey level observed during warming or cooling? These quantities should be defined in the caption and text. Is the scale 0-255 or 0-1 or 0-100? What is std mean on the y-axis? Are the curves the population mean, or are they for a single droplet? The thermodynamic prediction for the eutectic melting point for the NaCl and pure water should be added to the graph.

Figure 3. The C and H (presumably cooling and heating) should be explained in the caption. The Delta T is a temperature and should have units of K. The freezing temperature of pure water should be given here. The text states that the delta T is evaluated against an extrapolated temperature at equilibrium conditions. Does that mean that the

equilibrium conditions T for freezing and melting are not constant in the plot?

Section 3.2 provides statistics for the T50 for several devices and repeats for individual devices. However, no spectra are shown. How is the repeatability vis-a-vis early freeze events? The authors should show an overlay of the temperature spectra for all of these samples to convince the reader of the repeatability across the full range of temperatures.

Conclusions → homogenous should be homogeneous
* * *

---

## Referee Comment (RC2) · Anonymous Referee #2 · 3 Oct 2017

Review of: The Weizmann Supercooled Droplets Observation 2 (WISDOM) on a Microarray and application for ambient 3, Reicher et al.

This work attempts to build on the extensive past literature on droplet freezing assays, by leveraging microfluidic technologies. Several calibrations experiments are performed on the stage in an attempt to quantify the accuracy and precision of measurements of ice nucleation rates and ice active site densities. It is noteworthy that assessing the absolute values of these quantities is a particularly challenging endeavour, owing to the lack of 'primary standards' whose nucleation behaviours are known to a high degree of certainty. Indeed, past intercomparison studies have yielded considerable spreads in rates and ice active site densities for homogeneous nucleation and the heterogeneous nucleators, and deviations between instruments has not been satisfactorily addressed.

While the current work could definitely be of interest to the community of researchers interested in atmospheric ice nucleation processes, there are serious gaps in the paper which I feel need to be addressed prior to consideration for publication. The majority of these gaps revolve around the qualitative nature of the comparisons performed, and lack of error analysis on the rates and ice active site densities determined.

Major issues

- In figures 5, 7, 8 & 9, there are no vertical error bars presented to represent the uncertainty in the measurements. What is for example, the uncertainty stemming from the stochastic nature of nucleation in the homogeneous freezing experiments? What is the effects of the uncertainty stemming from the random sampling of ice nuclei into droplets on the heterogeneous ice active site densities in figures 7, 8, 9. Without statistically sound error bars and confidence intervals on the certainty of the data, comparisons are rendered almost meaningless.
- The authors note in the abstract (L16), that the method produces excellent 'statistics'. To what quantity are the authors referring to here? Nucleation rates? Ice active densities? If so, what is the effects of sampling hundreds of droplets over say 50? By what kind of factors are uncertainties reduced? If this is purported to be a major advantage of the technique, surely the improvement in these 'statistics' by this method should be quantified?
- in this direction, some of the benefits of this technique over past techniques should be elaborated upon and clarified; the discussion of past issues, without acknowledging how they have been successfully dealt with in the past is rather peculiar. For instance, many cold stage instruments do not suffer from issues surrounding the Bergeron-Findeison process, and do not need oil to be placed on the droplets. At line 53, it is said that generation of 1 uL volumes is not trivial. To my understanding, this can be done with a pipette, which would seem rather trivial to me.
- In section 2.3, on the automated detection of phase transitions, it is noted that the algorithm can 'sucessfully distinguish between a phase transition event and noise' (L 150). Whilst this statement may well be correct, I see no mention of to what accuracy the algorithm can successfully distinguish between phase transitions. Is this 100% accuracy? How many experiments were performed manually to determine this?
- In the quoted value of ±0.25 K for the Linkam cryostage temperature sensor, which is subsequently quoted in the captions for figures such as 5 and 7, how was this value

determined? Knowing this would certainly be useful for the reader. By what procedure was this value obtained?

- In figure 7, the data for NX illite appear to be at the extreme lower end of the spread, based on the error bars used for the Hiranuma data. Surely this should be discussed in the text?
- In lines 268-269, it is said that ns is in best agreement with the Leeds-NIPI for NX-illite. Yet close inspection (the subtle shades of grey used here in the graph make this a bit difficult to see), shows that there is in fact no overlap in the temperature range between the measurements presented here, and those of the Leeds NIPI uL.
- In addition, in comparison to the binary instrument in figure 7, the data are up to an order of magnitude or greater off, which is not immediately obvious as the authors have chosen to only label the scale for every factor of 100 increase. It is noted that this is within the uncertainty of the instruments, but what is the uncertainty of the quoted values for the WISDOM (see my first point above…).  Does the uncertainty really cover 2 orders of magnitude? With what degree of statistical certainty are you sure that these two measurements are in agreement?
- Lines 136-137: If the chips are being clogged by larger particles, then you may be severely altering the size-dependent particle composition of the samples as they pass through. How is this dealt with and accounted for?

Other issues

- Line 28: INP should be INPs
- Line 34: Why is this only 'possibly' in future climates?

---

## Author Response (AR1)

**Author's response to the reviewers and the revised paper:**

**Response to Reviewer #1**

**The authors are grateful to the reviewer for the important comments that helped them clarify the manuscript.**

**In black and bold are quotations that the reviewer cites from the manuscript.**

Reicher et al. describe the calibration and operation of a cold-stage for measurement of ice nucleation activity. The instrument used a microfluidic device for generation of monodisperse droplets inside an oil phase. The device is operated in batch mode. Droplets are prepared in a continuous flow system. Then the flow is stopped and the device is placed on a cold-stage/microscope for nucleation experiments. Characterization experiments using pure water, eutectic solutions, and several test dusts are described. Application of the technique to ambient aerosol is demonstrated. This is a solid paper that describes the application of a microfluidic device for ice nucleation studies. The characterization and validation experiments are of high quality. Specifically, the temperature characterization and comparison with the various test dusts at water activity of one and less than unity are thorough and convincing. The main conclusion of the paper that "WISDOM is a suitable tool for studying atmospheric ice nucleation, both in homogeneous and heterogeneous immersion freezing" is well justified. However, the paper should be viewed in perspective of the current literature. At least 10+ similar cold-stages have been built, characterized, and described in the last few years. This is also not the first cold-stage that uses a microfluidic device. In fact the predecessor work by Stan et al. (2009) managed to operate the instrument in continuous flow mode, rather than batch mode, and thus is more technologically advanced than the work here. In many places the manuscript tries to 'sell' or 'justify' the microfluidic technique presented here as an advance in technological capabilities. For examples, the microfluidic technique is 'cheap', 'easy to prepare', 'easy to operate', 'production of droplets is fast', 'a range of droplet volumes can be used', 'cooling down to homogenous temperature range allows an extensive investigation of atmospheric particles'. A list is provided that suggests that WISDOM solves 'some critical issues inherent in other currently used instruments', including 'fast production of droplets', high statistical power' due to 'fast analysis of thousands of droplets', 'droplets are monodisperse and individually analyzed', 'the use of oil minimizes possible artefacts', 'small droplet volumes decreases freezing artefacts by impurities, 'static array opens the possibility to investigate several freezing cycles for the same droplets', and 'microfluidics method and the small droplet volumes enable working with small volumes'. The statements above are either misleading or wrong or have not been demonstrated in this paper (see further below). It indicates that the authors have not critically reflected on the differences and similarity of WISDOM with the current technology and some of the necessary tradeoffs that are being made when designing cold stages. The WISDOM instrument does not demonstrate any aspects that has not been also addressed in other designs. A blunt assessment of the technique is that it is on par with the current state of cold-stage designs. However, although the technique is certainly valuable (and cool!), it does not represent an advance in either science or technology relative to the

published literature. A revised manuscript must better reflect the techniques ability and limitations relative to other existing techniques. Issues with justification. Quotes from the manuscript are in bold font

**Authors' Reply**: We have addressed the comments and re-wrote parts of the manuscript to adhere to the comments made by the referee. We believe that it now better reflects the abilities and limitations of WISDOM relative to other existing techniques.

**The microfluidic chips are cheap and easy to prepare and to operate. Production of droplets is fast and a range of droplet volumes can be used within the same microfluidic device.** These statements are highly subjective. Please elaborate. What does cheap mean? The chips are custom made, there is cost for that equipment and labor involved, which is not free. On top the system requires a microscope, liquid flow control, and more complex cooling setup relative to a regular cold-stage. What does fast mean? If cheap and fast are a 'selling' argument for the technique, then it should be compared to other cold-stage techniques. The real metric that need to be compared relative to other techniques. Total cost of equipment, total cost per 'experiment' in terms of equipment and consumables. (2) Number of drops and total volume that can be studied in a reasonable 'experiment' (e.g. working with a dust sample for one morning). Related to this number is the lowest and highest INP concentrations that can be captured in that experiment. Factors such as channel clogging, or chip damage due to expansion of water upon freezing should be accounted for here fairly. (3) Ease of use relative to a regular cold-stage technique. These are the key pieces of information that are needed to weigh whether one should adopt this technique or not (other than personal preference, which may well justify the route of microfluidics).

**Authors' Reply:** The descriptions "cheap and easy" referred to the microfluidic chips (or devices) and not to the entire WISDOM system. However, the authors understand the reviewer's concern and in order to avoid any subjective declarations this statement is removed from the manuscript.

In more detail:

(1)        The authors agree that a statement about the microfluidic system cost is subjective and will differ from lab to lab. However, in the manuscript we only claimed that the microfluidic devices themselves are cheap and do not refer to the entire system. Because cost is subjective, and we could not find this information in any other paper from the relevant fields, we could not provide a comparison of the costs with different cold stages. Hence, we removed this statement from the text, from the introduction and from the summary. We evaluate our devices at 1 euro per each device, the cost consists of the cost of a microscope glass slide and the cost of a PDMS layer on top. We do not expect this to change much from place to place.

(2)        Fast: Time to produce the droplets is about 30-40 seconds and then about 10 seconds to disconnect the device from the tubes and to transfer it to the cooling stage.

We cannot provide any specific information about clogging of the devices, as we did not study this in detail. Clogging will depend on the particles' properties and concentration and maybe on the flows used. Hence if channels are clogged or damaged (normally rare and we do recycle each device for a few experiments) we do not use the device since it is unknown how this

affects the concentration of the material inside the droplets or the change in the temperature equilibration in the device if the channels are damaged. In section 3.6 we state these issues where we discuss disadvantages of this system.

(3)      The authors agree here too, a statement about ease of use in comparison with other cold stages is subjective. In the quoted statement it was claimed that the preparation of the microfluidic device is easy and not that WISDOM is easier than the other cold stages.

The authors would like to clarify that the cooling stage is not more complex relative to the other stage-based instrument as the reviewer pointed out and similar cooling stages are in use within the ice nucleation community. We did not find in the text where it was presented as complex.

**A range of droplet volumes can be used** other cold-stages have worked with the same range of droplet volumes and sample statistics

**Authors' Reply:** The statement above introduces the versatility of the microfluidic device to study ice nucleation in droplets of varying volumes. For example, device with 100-μm diameter can be used to produce droplets with any diameter that is smaller than 100 microns.

**Cooling down to homogenous temperature range allows an extensive investigation of atmospheric particles Furthermore, supercooling is limited due to the presence of impurities, which increases with the volume of the droplet. Hence, to allow comprehensive studies down to the homogeneous region, low volumes are used and generation of these volumes is not trivial and may cause further complications.** The authors should elaborate on the point they are trying to convey here. Presumably, this is meant to convey that a technique like WISDOM is needed? Three points. First, may authors have accomplished studying homogeneous freezing using small drops successfully in past and present studies. Clearly it is doable and the alluded to 'complications' have been solved in some way by these authors. Second, one might be inclined to believe that going to small drop volume solves some problems. If done correctly, small volumes do allow generation of drops with no impurities and thus studying homogeneous nucleation (as demonstrated widely in the literature). However, the impurities are still present. Simply subdividing the sample into more droplets will not help raising the lower limit of detection of ice nucleation activity for that sample. Specifically, if a sample has 105 nuclei per liter of liquid, studying a large number of small droplets or a smaller number of large droplets will produce the same result. This study has not demonstrated any advance in purifying water. The achievement of pure water freezing has been managed before by others when working with small volumes. Third, the number of drops studied per experiment here is quite small (500). While this means that homogeneous freezing temperature can be reached, it also means that the total volume studied is quite small, and thus the lower limit of detection for ice nuclei is much larger than in devices that use larger droplets.

**Authors' Reply:** The authors made this statement to explain what the considerations to employ low volumes of droplets were.

(1) The authors did not claim that homogeneous freezing in low volumes was not studied successfully before. In fact, Figure 5 and Figure 6 present some of these studies which are used as benchmarks for the WISDOM studies.

(2) With all due respect, the authors do not agree here with the referee. How the active sites are distributed as a function of temperature is critical here. It is indeed correct that ice nucleation of materials can be studied well by using higher volumes of droplets, if the nucleation sites are active at warmer temperatures than the freezing of the impurities that are present in the sample (normally between ~-20 to -25C (Hiranuma et al. 2015, ACP)). A good example for this is *snomax*. In the atmosphere, supercooling is observed down to -37C and mineral dust have also active sites in colder regions than -25C. Therefore, low volumes are more sensitive to the active sites that are less rare and that are activated between -25C and down to the homogenous region. This was our consideration. In contrast, WISDOM is not sensitive to the rare active sites if they are active at warmer temperatures. In some cases, it is impossible to increase the surface area in the droplets (if there is not enough sample or if the high particle concentration disables the droplets production), then it will be possible to use a larger sample of droplets. This important point is now added to section 3.6:

*"…. the small droplets' volumes reduce the sensitivity to rare active sites. This may be solved by performing many experiments or by using larger droplets with more surface area within the droplets".*

(3) 500 droplets are normally not small for the concentrations and the materials that we use. The considerations regarding the effect of overall volume are given in (2) above.

**WISDOM solved some critical issues inherent in other currently used instruments including (1) fast production of droplets minimizes sample sedimentation or other aging process that may occur in a suspension, leading to higher reliability of the measurements and to a better estimation of the surface area of the material that is exposed in the droplets.** This has not been shown here. First, it is unclear how fast fast is. No times are given in the paper. Second, it has not been compared to how fast others can perform an experiment. Third, the paper does not demonstrate that this technique is more reliable than others. There is no metric for reliability. This statement is clearly not justified. (2) **The high statistical power that can be achieved easily by fast analysis of thousands of droplets.** Again, fast is subjective. The number of drops given here is that approximately 550 forty-microns and 120 hundred-micron droplets can be monitored per experiment. The duration of an experiment is unclear, but it includes chip production, chip loading, and post processing. So how many experiments can one person do in a week? Is the statistical power really higher than in other studies? For example, Hader et al. (2014, ACP) generate and analyze 500-800 drops per experiment in the 80-100-micron size range. Another example Peckhaus et al. (2016, ACP) generate and analyze 1200- 1500 drops drops per experiment om 100-micron size range. Both studies also use oil immersion. Does WISDOM have really faster analysis and higher statistical power than these studies? Perhaps so, but it must be proven.

**Authors' Reply:**

(1)      We agree that "fast" is subjective - removed from the text.

(2)      High statistical power - removed from the text.

**(3) the droplets are monodispersed and individually analysed, in contrast to other emulsion techniques** See above for examples that also use individual analysis.

**Authors' Reply:** In Hader et al (2014) the droplets are not monodispersed and in Peckhaus et al (2016) do not have an emulsion. The authors refer here to few other studies that used emulsion in the DSC for example and recorded the average freezing behaviour. This meant to clarify that there are some more parameters as frozen fraction and freezing rate that can be obtained with individual droplets, which cannot be obtained by bulk methods. We removed the sentence "in contrast with other emulsion techniques".

**(4) the use of oil minimizes possible artefacts from droplets' evaporation, neighbour seeding or vapour** . . . See above cited studies and several other systems that do the same.

**Authors' Reply:** As was explained earlier, these are advantages in WISDOM that the authors find important and maybe this adds to the variability between the different techniques.  It is important to clarify that it is not stated that WISDOM is the only cold stage setup for ice nucleation studies that has these properties.

**(5) the small droplet volumes decrease freezing artefacts by impurities, and in the absence of INPs the water freezes below the homogenous freezing threshold (-37°C), in comparison with instruments that employ droplets with larger volumes which limit the workable temperature range.** This point is incorrect as stated above. Working with small droplet volumes allows studying dilute solutions and ice nuclei that have high concentration in liquid. The high concentrations are the result of the overall low liquid volume that is analysed. Working with large droplets is a choice to increase the detection limit. Furthermore, many investigators work with small droplets before, so this is not a distinguishing feature of WISDOM.

**Authors' Reply:** (5) Similar points are explained before.

**(6) using a static array opens the possibility to investigate several freezing cycles for the same droplets** Re-freeze experiments have been performed with static droplet arrays since the pioneering work by Vali in the 1960's. This is not a distinguishing feature of WISDOM.

**Authors' Reply:** (6) the authors did not claim that this is a distinguishing feature of WISDOM. The text only describes other possible experiments in WISDOM.

**(7) the microfluidics method and the small droplet volumes enables working with small volumes which is an advantage when working with atmospheric samples.** First, working with small volumes and atmospheric samples has been also done by previous investigators. Second, it is incorrect to cite the small droplet volume as an advantage. It may be an advantage under some circumstances, but not others. Specifically, the poor lower limit of detection for small droplets is a clear disadvantage. To summarize the critique of this section. The text is prefaced with "WISDOM's largest advantage is the use of microfluidics technology which solves some critical issues inherent in other currently used instruments". The issues that WISDOM solves have been solved in some form or another in previous studies. WISDOM shares the advantages and disadvantages of the design choices involving the selection of oil, droplet size, batch mode, and optical detection. Claims regarding high statistical power, ease of use, and cost need to be proven. The paper should give an honest assessment of whether a new investigator should pursue the microfluidics route or the setup or something else. WISDOM and the nanoliter droplet freezing assay have similar control over drop volume, automated drop generation, drop detection, refreeze capabilities, etc. Perhaps telling is that the entire section in the draft manuscript is devoid of any references to the literature to back up their claims.

**Authors' Reply:** (7) we explained the considerations for choosing the small volume in the previous replies. The authors do not claim that it was not done before, and we clearly discuss the advantages and disadvantages of the volume range chosen. In any case, microfluidics allows the use of higher volumes. The authors of this manuscript made every effort to describe the system in an objective way, and to cover existing literature.

Other comments

**However, in the atmospheric heterogeneous ice nucleation field, microfluidics techniques are not widely adopted, despite many potential advantages.**

What are these potential advantages? Please be specific. Also, there are clearly potential disadvantages to microfluidics. These should be discussed as well.

**Authors' Reply:** These points are clearly stated in section 3.6, the authors corrected section 3.6 in order to represent better the advantages and disadvantages of WISDOM, considering the points that the reviewer raised:

*"WISDOM uses of microfluidics technology which solves some critical issues inherent in other currently used instruments: (1) good control of size number of monodisperse droplets, (2) fast production of hundreds of nearly monodisperse droplets minimizes sample sedimentation or agglomeration that may occur in a suspension, leading to a good estimation of the surface area of the suspended material. Moreover, several droplet diameters can be employed in the same device without its modification, (3) good statistics achieved by individual analysis of thousands of droplets, (4) monodisperse droplets individually analyzed, in contrast to some emulsion techniques as the Differential Scanning Calorimeter (DSC) experiments, allow to obtain the frozen fraction at each temperature, and to achieve detailed information about active sites*

*and freezing rates, (5) the use of oil minimizes possible artefacts from droplets' evaporation, neighbor seeding or vapor transfer due to the Wegener–Bergeron–Findeisen processes, (6) the small droplets' volume decreases freezing artefacts by impurities, allows to reach the homogenous freezing threshold (-37ºC), (7) possible investigation of several freezing cycles for the same droplets, (8) the microfluidics method and the small droplet volumes enable working with small sample*
5 *volumes which can be an advantage when working with atmospheric samples.*

*WISDOM has a very accurate temperature calibration that spans a wide temperature range, using the eutectic freezing method. WISDOM most resembles the instrument used by in Edd et al. (2009). However, it seems that issues with temperature calibration in Edd et al. (2009) led to a temperature offset, and hence different freezing rates. Stan et al. (2009) achieved better temperature accuracy and high statistics. However, the freezing experiment was conducted in a flow mode,*
10 *which is more complicated than in the WISDOM setup and requires complicated modeling. In addition, the cooling rates that were used were very fast, which induces additional errors. Riechers et al. (2013) had high temperature accuracy as they also used a DSC. However, they had to collect the droplets from the device as there was no static array option and this may add further complication and contamination.*

*It is noted also that use of microfluidics technology has a few disadvantages. These may include: (1) oil may interact with*
15 *some of the analyzed particles, possibly leading to biased data, (2) the microchannels are susceptible to clogging, (3) it is not possible to perform any post analysis to the droplets content after the experiment, (4) because of the small droplets' volumes, and there is less chance to characterize rare active sites".*

Figure 2 needs better explanation. The y-axis is not clearly defined. Presumably, Delta GL stands for change in grey level
20 observed during warming or cooling? These quantities should be defined in the caption and text. Is the scale 0-255 or 0-1 or 0-100? What is std mean on the y-axis? Are the curves the population mean, or are they for a single droplet? The thermodynamic prediction for the eutectic melting point for the NaCl and pure water should be added to the graph.

**Authors' Reply:** More detailed explanation is added to the text in section 2.3:
25 *"The optical brightness of a droplet changes during a phase transition (freezing or melting) due to the different interaction of light with the liquid and the solids. For phase transition detection, an in-house image processing LabVIEW program monitors automatically the optical brightness change. The program detects the droplets using a spherical shape criterion and sets a square surrounding the droplet that defines an array of pixels that are attributed to that specific droplet. A change in the optical brightness is represented by the gray level value of the image's pixels, ranging from 0 to 255. Freezing is*
30 *calculated per movie frame and is defined as the subtraction of the brightness mean value for each droplet in two consecutive frames ($\Delta GL$), thus allowing derivation of freezing rates. At the beginning of the analysis, the first 15 frames are used to identify the noise level of the signal by calculating its standard deviation (std($\Delta GL$)). The program then searches for the maximal freezing signal that is also greater than 5 times the noise level. The temperature associated with this freezing signal is assigned as the freezing temperature for that droplet.*

*In this algorithm, the program can distinguish successfully between a phase transition event and noise that arises from the camera signal, droplet movement or any other interruption. Figure 2 presents a spectral analysis for different types of phase transitions observed in WISDOM. Since WISDOM operates in transmission microscopy mode, the light is scattered more efficiently by ice crystals in comparison with a liquid droplet and a freezing event involves droplet darkening and a negative signal. Example for the negative signal of a freezing event of a single droplet can be seen in Figure 2a. In comparison, during melting, the droplet becomes brighter until all the crystals melt, and the signal is positive. In Figure 2b+c the analysis of melting signal and eutectic melting signal are presented for the whole frame".*

And to the figure 2 caption:

*"Spectra of different phase transition events as observed in WISDOM. a) freezing, b) eutectic melting, and c) melting onset and clear point (liquefaction) are the mean of all sampled droplets in a single experiment. The phase transition is defined optically by the brightness information obtained by the gray level of the image pixels. Std(ΔGL) describes the standard error of the difference in mean GL for two consecutive frames. At the beginning of the experiment the noise level is studied and freezing or melting is detected only if std(ΔGL) is as least 5 times greater than the noise std level. Freezing and melting examples are for pure water droplets and the eutectic melting example is for aqueous solution droplets of NaCl. Eutectic melting point of NaCl and pure water melting point are marked by the yellow and red lines in b and c, correspondingly. In all cases the droplets diameter is 100 μm."*

melting points were added to the graph as well.

Figure 3. The C and H (presumably cooling and heating) should be explained in the caption. The Delta T is a temperature and should have units of K. The freezing temperature of pure water should be given here. The text states that the delta T is evaluated against an extrapolated temperature at equilibrium conditions. Does that mean that the equilibrium conditions T for freezing and melting are not constant in the plot?

**Authors' Reply:** Explanation is added to figures 3. The extrapolation results in equilibration are the values that we calibrate against.

Section 3.2 provides statistics for the T50 for several devices and repeats for individual devices. However, no spectra are shown. How is the repeatability vis-a-vis early freeze events? The authors should show an overlay of the temperature spectra for all of these samples to convince the reader of the repeatability across the full range of temperatures.

**Authors' Reply:** Per the Reviewer's request, a graph of the full range is now added in appendix A.

Conclusions → homogenous should be homogeneous. **Authors' Reply:** Corrected in the text, thank you!

**Response to Reviewer #2**

**The authors appreciate the thorough review and would like to thank the reviewer for taking the time and for helping to improve our manuscript, especially it's qualitative nature.**

This work attempts to build on the extensive past literature on droplet freezing assays, by leveraging microfluidic technologies. Several calibrations experiments are performed on the stage in an attempt to quantify the accuracy and precision of measurements of ice nucleation rates and ice active site densities. It is noteworthy that assessing the absolute values of these quantities is a particularly challenging endeavor, owing to the lack of 'primary standards' whose nucleation behaviours are known to a high degree of certainty. Indeed, past intercomparing studies have yielded considerable spreads in rates and ice active site densities for homogeneous nucleation and the heterogeneous nucleators, and deviations between instruments has not been satisfactorily addressed. While the current work could definitely be of interest to the community of researchers interested in atmospheric ice nucleation processes, there are serious gaps in the paper which I feel need to be addressed prior to consideration for publication. The majority of these gaps revolve around the qualitative nature of the comparisons performed, and lack of error analysis on the rates and ice active site densities determined.

Major issues

• In figures 5, 7, 8 & 9, there are no vertical error bars presented to represent the uncertainty in the measurements. What is for example, the uncertainty stemming from the stochastic nature of nucleation in the homogeneous freezing experiments? What is the effects of the uncertainty stemming from the random sampling of ice nuclei into droplets on the heterogeneous ice active site densities in figures 7, 8, 9. Without statistically sound error bars and confidence intervals on the certainty of the data, comparisons are rendered almost meaningless.

**Authors' Reply:** In figure 5 error bars have been added for WISDOM and for the rest of the used data. In figure 7 the errors were confined in the marker itself, both for x and y axis. We preferred different marker sizes because visually it was hard to place the error bars in a clear way. This is now mentioned in the figure caption:

*"Accumulated active site density spectra ($n_s$) of K-feldspar and Illite-NX particles as a function of temperature from validation experiments of immersion freezing in WISDOM. Frozen fraction values are represented by a color bar, for few surface area values that are exposed in 40 and 100 μm droplets. The dependence of the nucleation site density on the surface area is illustrated here. WISDOM uncertainties are included within the size of the markers. The uncertainty in $n_s$ is propagated from the error in the surface area and the error in the frozen fraction. For validation, previous immersion freezing measurements are also presented (Hiranuma et al. (2015) and Atkinson et al. (2013)). For the Hiranuma et. al. fit, the maximum deviation between maxima and minima in the vertical axis are shown by the error bars for the relevant temperature range."*

Moreover, new graphs with the error bars for the illite and for the k-feldspar data is now added in appendix B.

**Authors' Reply:** In figure 8 and 9 error bars are added.

5 • The authors note in the abstract (L16), that the method produces excellent 'statistics'. To what quantity are the authors referring to here? Nucleation rates? Ice active densities? If so, what is the effects of sampling hundreds of droplets over say 50? By what kind of factors are uncertainties reduced? If this is purported to be a major advantage of the technique, surely the improvement in these 'statistics' by this method should be quantified?

10 **Authors' Reply:** Due to the concern of both reviewers regarding the term "excellent statistics", this was removed from the manuscript:

*"Frozen fraction, ice nucleation active surface site (INAS) densities and freezing kinetics can be obtained from WISDOM measurements using hundreds of individual droplets in a single freezing experiment. Extensive calibration experiments using eutectic solutions and previously studied materials are described".*

• in this direction, some of the benefits of this technique over past techniques should be elaborated upon and clarified; the discussion of past issues, without acknowledging how they have been successfully dealt with in the past is rather peculiar. For instance, many cold stage instruments do not suffer from issues surrounding the Bergeron-Findeison process, and do not need oil to be placed on the droplets. At line 53, it is said that generation of 1 uL volumes is not trivial. To my understanding, this 20 can be done with a pipette, which would seem rather trivial to me.

**Authors' Reply:** In section 3.6 the advantages and disadvantages of the WISDOM are discussed. Also, the considerations for choosing the microfluidics approach are described. We further show the importance of a thorough calibration of temperature and temperature equilibration properties. Clarification: in line 53, it is claimed that smaller volumes are not trivial to generate 25 and not 1 microliter droplets, which is indeed trivial.

• In section 2.3, on the automated detection of phase transitions, it is noted that the algorithm can 'successfully distinguish between a phase transition event and noise' (L 150). Whilst this statement may well be correct, I see no mention of to what accuracy the algorithm can successfully distinguish between phase transitions. Is this 100% accuracy? How many experiments 30 were performed manually to determine this?

**Authors' Reply:** This is monitored for each freezing event and done separately for each experiment since the noise level varies between experiments. The freezing causes larger reduction of the grey level than the noise, and hence it is possible to

separate it from the noise. So, a threshold is being set to differentiate the noise. This leads to very high success rate in identifying the freezing events.

• In the quoted value of ±0.25 K for the Linkam cryostage temperature sensor, which is subsequently quoted in the captions for figures such as 5 and 7, how was this value determined? Knowing this would certainly be useful for the reader. By what procedure was this value obtained?

**Authors' Reply:** This value ($<\pm0.25K$) is the uncertainty of the Pt100 temperature sensor in the temperature range. Using this value with our calibration results we propagate the total temperature uncertainty ($\pm0.3K$). This was not clear in the text and we thank the Reviewer for this comment which is now better explained in the text.

• In figure 7, the data for NX illite appear to be at the extreme lower end of the spread, based on the error bars used for the Hiranuma et al. data. Surely this should be discussed in the text?

**Authors' Reply:** The comprehensive data of Hiranuma et al. covers different instruments and several immersion freezing techniques. Hiranuma et al. also mention the spread of the results, up to three orders of magnitude. In the effective WISDOM's temperature range, the fit of the results from only two instruments (CU-RMCS and NC-State-CS) that are relatively different from each other, in comparison to the spread among most of the other instruments. WISDOM and CU-RMCS agree well within the same order of magnitude on the number of active sites. Investigating the reasons for the spread in the results of all instrumentations is beyond the scope of this paper and is thoroughly discussed in Hiranuma et al. The Binary and Leeds data were also fixed to $n_s$ that is based on BET measurement, instead of the data that was $n_s$ based on geometry. Our results are also normalised to the BET surface area and now the agreement with Binary and Leeds are clearer.

• In lines 268-269, it is said that ns is in best agreement with the Leeds-NIPI for NX-illite. Yet close inspection (the subtle shades of grey used here in the graph make this a bit difficult to see), shows that there is in fact no overlap in the temperature range between the measurements presented here, and those of the Leeds NIPI μL.

**Authors' Reply:** Explanation is united with the reply to the next comment.

• In addition, in comparison to the binary instrument in figure 7, the data are up to an order of magnitude or greater off, which is not immediately obvious as the authors have chosen to only label the scale for every factor of 100 increase. It is noted that this is within the uncertainty of the instruments, but what is the uncertainty of the quoted values for the WISDOM (see my first point above…). Does the uncertainty really cover 2 orders of magnitude? With what degree of statistical certainty are you sure that these two measurements are in agreement?

**Authors' Reply:** The conclusion that the Illite-NX data is in good agreement with the BINARY rely on the fact that there is less than one order of magnitude between the two data sets and this difference is justified by uncertainties of both instruments. The temperature at which there is more than an order of magnitude difference (~250K) is where the variation between our measurements are the highest and for < 0.01% of the droplets. This is where very rare active sites exist (in the used surface area) and is less representative of the material. Moreover, if the slope of the Leeds NIPI is extrapolated, it would nicely overlap with WISDOM data. Here it is also expected that the data is of higher significance as there are many more freezing events. Examination of the Hiranuma et al data and our data, it seems that the data converge to the WISDOM and CU-RMCS results and slope:

[Figure]

*The labelling of the y axis is changed to every factor of 10 increase to avoid confusion of the readers.

*The figure is now replaced with a new figure that contains data from several instruments from Hiranuma et al. (2015) and the data from different instruments is now represented by colours instead of the grey shades.

• Lines 136-137: If the chips are being clogged by larger particles, then you may be severely altering the size-dependent particle composition of the samples as they pass through. How is this dealt with and accounted for?

**Authors' Reply:** This is indeed a concern and we do not work with clogged devices or with materials that tend to clog the devices. Moreover, we repeat experiments with the same device for few times (with fresh droplets), and do not observe evidence of critical alteration in the size distribution. We would expect that such alterations will increase with increasing number of repetitions and that we will see a reduction in the ice formation efficiency. For materials with particles >5 micron

in diameter, we experienced immediate clogging. From our experience, these particles readily settle in the suspension, even before the droplet production and also in cases where it was stirred during the droplets production. We also expect that for materials that tend to clog the device, it will be difficult to quantify that as it might change with the flow used or concentration of the material in the suspension and the material's size distributions. Hence, we do not work with suspensions of larger particles or when we observe evidence for clogging.

Other issues

• Line 28: INP should be INPs

INP changed in the text to INPS. Thank you!

• Line 34: Why is this only 'possibly' in future climates?

How can we be certain that this will affect climate?

We have performed further changes in the revised version of the manuscript:

- The Illite-NX $n_s$ values used in figure 7, were $n_s$-geo instead of ns-BET, and now it is corrected.

- The authors have realized that the $n_s$ values given in figures 8+9 is over estimation of $n_s$ due to a mistake in the surface area estimation. The graph is now updated with the correct values. The revised manuscript contains examples of three MOUDI's stages, collected in one intensive Saharan dust event, that were comprise of a very small amount of dust.

Changes in the manuscript are highlighted in yellow, and sentences that were cut are crossed with a line.

[revised manuscript text omitted]